# Successional Categorization of European Hemi-boreal Forest Tree Species

**DOI:** 10.3390/plants9101381

**Published:** 2020-10-16

**Authors:** Raimundas Petrokas, Virgilijus Baliuckas, Michael Manton

**Affiliations:** 1Department of Forest Genetics and Tree Breeding, Institute of Forestry, Lithuanian Research Centre for Agriculture and Forestry, Kaunas distr LT-53101, Lithuania; raimundas.petrokas@lammc.lt (R.P.); virgilijus.baliuckas@lammc.lt (V.B.); 2Institute of Forest Biology and Silviculture, Vytautas Magnus University, Studentu 11, Akademija, Kaunas LT-53361, Lithuania

**Keywords:** climatic climax, life history, forest disturbance, gap colonizers, gap competitors, forest colonizers, forest competitors, forest dynamics, forest management

## Abstract

Developing forest harvesting regimes that mimic natural forest dynamics requires knowledge on typical species behaviors and how they respond to environmental conditions. Species regeneration and survival after disturbance depends on a species’ life history traits. Therefore, forest succession determines the extent to which forest communities are able to cope with environmental change. The aim of this review was to (i) review the life history dynamics of hemi-boreal tree species in the context of ecological succession, and (ii) categorize each of these tree species into one of four successional development groups (gap colonizers, gap competitors, forest colonizers, or forest competitors). To do this we embraced the super-organism approach to plant communities using their life history dynamics and traits. Our review touches on the importance and vulnerability of these four types of successional groups, their absence and presence in the community, and how they can be used as a core component to evaluate if the development of the community is progressing towards the restoration of the climatic climax. Applying a theoretical framework to generate ideas, we suggest that forests should be managed to maintain environmental conditions that support the natural variety and sequence of tree species’ life histories by promoting genetic invariance and to help secure ecosystem resilience for the future. This could be achieved by employing harvesting methods that emulate natural disturbances and regeneration programs that contribute to maintenance of the four successional groups.

## 1. Background

Forests are complex systems of interacting organisms; to manage them for tree species composition and production we need thorough knowledge of the variety of tree species’ life histories and how they interact. Within the hemi-boreal forest climatic zone there are three main forest disturbance regimes that host a variety of successional characteristics: (i) stand succession (large or stand replacing disturbance such as severe fire, windthrows, or current clear felling), (ii) cohort dynamics (related to partial disturbances of a stand such as a low intensity ground fire or forest thinning), and (iii) gap dynamics (such as small patch or a fallen tree) [1]. Succession is a sequential shift of patterns and processes in terms of the relative abundance of dominant species [2]. The succession of forest stands and patches largely determines the extent to which forest communities are able to cope with changes in environmental conditions and forest loss due to natural disturbances or human activity [3,4,5,6]. Forest disturbances trigger successional events that lead to climatically determined end communities or climatic climax, generally regarded as a position of stability in the development of vegetation [7,8,9]. 

The first definition of climax was described by Clements [10] as the ability of species composition to remains stable for more than one tree generation (i.e., the tree species replace themselves) in the absence of disturbance other than tree deaths due to old age. Thus, a forest that can regenerate naturally with the same composition over time can qualify as natural climax. Although Clements’ [10] dynamic ecology concept is still valid [11], it does not represent the boundless factors impacting ecological succession. For example, the role and importance of both biotic and abiotic factors in predicting species distributions remains unclear [12,13,14,15,16]. Therefore, no clear conclusion can be drawn as to the successional position of tree species [10]. The probability of species survival and succession after disturbance depends on a species’ genetic profile to deal with a variety of environmental characteristics [17]. In other words, a tree’s life history traits define its position along its successional pathway that includes functional strategies for reproduction or resource capture [18]. 

The fundamental principle underlying the theory of invariance is that the laws of nature always have the same form for all observers [19]. This implies that all the elements of any developing living system interact, and thus all elements are ecologically equivalent, as the essence of ecological law and processes lies in invariance by which a living system following a disturbance returns to its stable state [20,21]. From a wildlife perspective, each organism, population, and community have different environmental scales in both time and space [22], and individual species may impact another species’ life history traits [23,24]. Thus, there are different perceptions about the interactions among species (that otherwise can survive virtually the same for millions of years), which proceed towards the ecological equivalence of climatically determined end communities [25]. Primary forests exist in a delicate but stable climax with all other components of the ecosystem; not one component can change without compensating changes in the others. For example, harvesting or thinning a forest stand will inevitably be followed by changes in the soil profile, vegetation, and life occurrence [9]. Generally, the dynamics of forest communities can be controlled by a set of ecologically invariant life-history traits of tree species turnovers [9,26,27,28,29].

The natural tendency of forest succession is towards climatic climax, whereas the succession of forests after human activity (e.g., fire, grazing, and soil deterioration due to over-cultivation) can result in adaptation of biotic climaxes [9]. Therefore, forest restoration that aids the recovery of forest structure, ecological functioning, and biodiversity towards those typical of a climax forest by the re-instatement of ecological processes is needed [30]. From an organism-centered perspective, developing forest management and exploitation regimes that mimic the natural conditions as closely as possible requires the determination of the degree to which typical species behaviors are responsible for the emergence of climatic climax [31,32,33,34,35,36].

The aim of this review was two-fold: (i) to review the life history dynamics of hemi-boreal tree species found in Lithuania in the context of ecological succession, and (ii) to categorize Lithuania’s forest tree species into four successional development classes. Finally, we discuss how they can be used to evaluate if the development of the community is progressing towards the restoration of the climatic climax. 

## 2. Successional Categorization of Forest Tree Species in Lithuania

Lithuania (62,000 km^2^) is situated in the hemi-boreal climatic zone (i.e., the transitional zone from temperate to boreal forests) and is affected by the humid marine climate of the Baltic Sea [37]. The natural potential forest cover of Lithuania is predominantly composed of five main forest types: (i) hemi-boreal spruce forest with mixed broadleaved trees (55%), (ii) mixed oak–hornbeam forests (22%), (iii) boreal and hemi-boreal pine forests with partial broadleaved trees (18%), (iv) lime-pedunculate oak forests (4%), and (v) species-poor oak and mixed oak forests (1%) [38]. Thus, the natural climatic climax of the region for tree species consisted of Scots pine (*Pinus sylvestris* L.), Norway spruce (*Picea abies* L.) Karst), birch (*Betula pendula* Roth and *B. pubescens* Ehrh), alder (*Alnus glutinosa* L. Gaertn. and *A. incana* L. Moench), English oak (*Quercus robur* L.), small-leaved lime (*Tilia cordata* Mill.), and European hornbeam (*Carpinus betulus* L.) [39]. Currently approximately 33% of Lithuania is forested with Scots pine, Norway spruce, and birch forming the dominating forest stand types [40]. The full range of hemi-boreal forest species found in Lithuania and their life history dynamics can be found in Table 1. 

## 3. Four Types of Forest Successional Groups for Lithuania

In landscape ecology, invariance of scale is a main concept that focuses on the influence exerted by spatio-temporal patterns of species distribution on the organization of, and interaction among, functionally integrated multispecies ecosystems [45,46]. Successional trajectories are an expression of sensitivity of interaction among the patterns of species distribution [35,45]. Watt [47] was a pioneer in linking space and time using the scale of landscape by applying two main forest dynamic models: (i) the patch–mosaic model, and (ii) the gap–phase model [45,48]. However, Watt [47] highlighted that phases were synonymous with patches, and that the gap phase could have different spatial dimensions, which could be considered as a stage of forest development where regeneration was confined. Nonetheless, patches within the landscape may not be self-evident, as environmental characteristics of different species, genotypes, or phenotypes may vary in scale [49]. For example, from a forest management and exploitation perspective, a patch may correspond to a forest stand, which may not function as a patch of habitat from a particular organism’s perspective [50]. Moreover, Bugmann’s [51] forest gap model analysis was not able confirm the assumption that a forest can be summarized as a composite of many horizontally homogeneous patches of land, where each patch can have a different age and successional stage, and that the successional processes may be described for each patch separately. Gap formation drives both the forest growth cycle and also determines forest floristics [52]. For instance, pioneer species often have a variety of specific life traits and in turn can produce sporadic and opportunistic patches during later stages of the succession [9]. In general, shade-tolerant species have a greater chance of successful regeneration in small gap openings compared to light-demanding species, which require larger gaps or stand replacement succession [49]. This is attributable to the contrasting growth patterns and life history traits of tree species with early-successional or large gap specialists exhibiting a height growth type, while on the other hand, late-successional or small gap specialists exhibit a crown growth type [49].

The trajectories of succession can be defined by rates of recruitment, growth, and mortality of a tree species population [53]. Therefore, using life history dynamic traits (e.g., natural regeneration dynamics, establishment, and growth (Table 1 and Appendix A), we categorized each of Lithuania’s forest tree species into one of the four types of successional groups based on environmental specialization of species and tree regeneration modes in forest gaps (Table 2). These resemble Clark and Clark’s [54] four dominant microsite patterns of tree species (A–D), which suggest significantly different regeneration biology, Whitmore’s [55] pioneer tree species index (1–4), Yamamoto’s [56] four major types of tree regeneration mode in gaps, Petrere et al.’s [36] four community types, and Chazdon et al.’s [53] old-growth generalists, successional generalists, and successional specialists.

## 4. General Suggestions for Forest Management

Many of the difficulties encountered by silviculturists and forest managers to secure natural regeneration of climax forest species occur due to the selection of shade-intolerant, rapid-growing, pioneer species or through the premature stopping of natural succession in order to maintain a forest in a certain seral stage [7]. However, large scale biogeography and climatic factors have been connected to the shade tolerance of tree species during regeneration succession [62], and the low light survival/high light growth trade-off is only one of the many possible characteristics of tree regeneration along a disturbance gradient [63]. 

The transition to post-disturbance stands dominated by fast growing shade intolerant tree species will eventually be replaced by late-seral shade-tolerant species. This is not a simple unidirectional sequence of stages, but rather a complex paradigm that is subject to responses of both species and environmental factors [45,64]. For example, Norway spruce is considered a climatic climax forest species in some parts of central Europe but is considered a pioneer species in northern Russia [7,65], or a climax beech forest after one generation may transit into a silver fir dominated stand in one area and a spruce dominated stand in another area [9]. Thus, understanding the dynamics of a tree community following a disturbance requires knowledge of the responses of individual tree species’ traits and how they interact within the local forest community [17]. The life history traits and strategies of individual species are intrinsically related to forest disturbances and account for the interaction among the patterns of species distribution [66]. This offers new ways to test the efficacy of specific interventions by modifying disturbance-related changes in dynamic forest communities [7,32,45]. 

A natural forest community, according to the theoretical framework applied in this study on successional categorization of the life history dynamics tree species, deems that forests should contain a mixture of tree species from each of the successional categories. This would ensure that the life history traits of both optimal colonizing tree species and optimal competitor tree species would be maintained. For optimal colonizing tree species, this would include dispersal, large somatic plasticity, and regeneration establishment dependency [7,67,68,69,70], whereas for optimal competitor tree species, this would include selective dispersal [71], large hereditary plasticity [72,73,74,75], and regeneration growth dependency [76]. Moreover, the genetic profile of predominantly self-fertilizing colonizing species indicates that there is cooperation between cross-pollinators and the population stability of the self-fertilizers, and this stimulates the adaptation towards a forest mosaic of small patches [7]. However, the interspecific interactions of new species assemblages provide both opportunities and challenges for species survival. Thus, a variety of tree species’ life histories and how they are integrated into the forest system need to be summarized as a continuum of ecologically invariant life-history trajectories of species turnover [77]. Unfortunately, there is limited research evidence as to why forests may lack certain types of successional pathways for species that have life history strategies that are needed to achieve natural climax. 

The abandonment of natural regeneration to focus on high yield forest management and timber production [7] is often still considered as a criterion of good forest management. However, how can forest managers, conservationists, and researchers help forest ecosystems maintain optimum stability? One avenue is to concentrate efforts to undertake vulnerability and risk assessments that feed into action plans toward managing native tree species [26]. For example, in the United States of America, the Forest Tree Genetic Risk Assessment System (ForGRAS) is used to rank forest tree species for a number of primary risk factors including population structure, rarity, regeneration capacity, dispersal ability, habitat affinity, genetic variation, pest and pathogen threats, and climate change pressure [4,33]. In general, these factors affect ecological processes that are important for ecosystem functioning, such as primary productivity, population recovery from disturbances, interspecific competition, community structure, and fluxes of energy and nutrients. Ultimately, a key management target is to conserve the genetic legacies for ecosystem memory and the adaptive ability they provide [78].

Indeed, one of the central paradigms in the biological theory is the idea that life adapts genetically to environmental change [33]. However, the available paleodata provides an independent testimony that such an adaptation does not take place for species that succeed each other in the paleorecord, instead they emerge unexpectedly as discrete morphological and genetic entities and survive virtually the same for millions of years [31]. Living ecosystems themselves represent a unique genetic mechanism responsible for the maintenance of Earth’s habitability. A lack of biotic regulation of environmental conditions would trigger large scale uncontrolled changes that would be four orders of magnitude faster than physicochemical processes [33]. For this reason, the variety of tree species’ life history traits and how they interact, constantly evolving toward the climatically determined end communities or climatic climax, is a manifestation of already existing genetic information, written in the genomes of species, which has remained practically unchanged for time periods in the order of several million years (mean time of species existence). Whenever the environment deviates from the optimum, genetically programmed species-typical behaviors and responses ensure that biotic processes can compensate for unfavorable change. The model of initial floristic composition postulates that most late successional tree species will become dominant due to an established soil seed bank or seedling bank [79]. However, in reality, the difference between a successional forest and a climax forest is subjective, as a forest ecosystem is dynamic, where succession is a continual process [32].

In the absence of external disturbances, a climax community is able to keep its own optimal environment stable for infinitely long periods of time [35,77]. For instance, many of the rarest hemi-boreal forest species are associated with ancient trees that still remain and can be tracked through time to a continuous cover of old trees [38,39]. Thus, understanding the biological legacies produced by natural disturbances and succession is crucial towards reaching sustainable forest management for both conservation and wood production [80].

Remnant tree species of the natural potential forests [38] continue to exist in Lithuania through the current dominant and secondary tree species, as outlined in Table 1. Large scale changes in forest cover, regeneration, establishment, species survival, and composition throughout millennia have led to biotic climaxes. For instance, natural oak, ash, and lime forests have become rarer due to both the past and current forest management objectives as well as altered hydrology. Over the past century, Lithuanian forestry has been juxtaposed between two contrasting approaches to forest management: German influence that favored artificial regeneration, and Russian reliance on natural regeneration after timber harvest [66]. Since regaining independence in 1991, Lithuanian forestry has turned into a timber production industry by employing silvicultural practices to generate sustained high yield wood production [40]. This is achieved by forming productive man-made forests that comply with site type conditions [40,81] and reduced harvesting ages of forest stands in comparison to natural forest succession (Table 1). As observed by Roberge and Angelstam [82] the anthropogenic impact on Lithuanian forest biodiversity is still lower compared to countries in western Europe. However, under Lithuania’s current forest management trajectory, it is expected that Lithuania’s forest biodiversity will follow the declining trends of western Europe [83]. In addition, European beech may be expanding its range into the Baltics through the introduction of forestry [84], and wild cherry is spreading into forest stands naturally from domesticated sources [57,60].

The experimental approach to climatic climax in Lithuanian forests until now had not been analyzed. Based on this review, we suggest that forest management should (i) utilize various harvesting methods that not only consider site type regeneration but also emulate natural disturbances, which includes but is not limited to clear cutting of stands, continuous cover forestry, and selective harvesting or gap creation [85]; (ii) the regeneration for forest stands should move away from single species regeneration by implementing the maintenance of the four types of successional groups (Table 2); and (iii) increase the felling ages by considering natural succession. As we have highlighted, this will enable the life history traits of the native forest tree species to evolve by invoking genetic invariance and to help secure ecosystem resilience for the future.

## 5. Concluding Remarks

This review focused on the life history dynamics (natural regeneration, establishment, growth, and survival) of 8 dominate tree species, 12 secondary tree species, and 4 introduced species for Lithuania’s hemi-boreal forests. Considering the dendroflora characteristics of Lithuania [41,42], we reviewed each tree species’ characteristics (Table 1 and Appendix A) and assigned each tree to one of the four successional categories (Table 2). This was done by embracing the super-organism approach to plant communities that succession is a universal process of a series of events where the types of vegetation in an area are directly related to climate [10]. This analysis of the successional categorization of hemi-boreal forest tree species exemplifies aspects of tree species’ life histories and how they can interact. All individual populations that form a continuous cover of trees throughout time on a site exhibit sensitive interactions. This sensitivity relates to the life history dynamics of tree species turnover towards the restoration of the climatic climax. The sensitivity of interaction among the patterns of species distribution provides new opportunities to test the efficacy of specific interventions to modify the disturbance-related changes in forest dynamics.

Due to the adverse risk and effects of climate change on forest and wildlife management, the organization and interaction among multi-species ecosystems need to be understood to forecast the dynamics of local ecological communities following disturbance. This would enable solutions that support climatic climax by introducing a successional approach to restore ecological processes, which would accelerate the recovery of forest structure, ecological functioning, and biodiversity levels towards those typical of climax forests based on predictive models for natural regeneration establishment, growth, and survival. The vulnerability of forest communities to negotiate anthropogenic impacts and climatic changes (the two major forms of disturbance occurring today) could be inferred by identifying the types of successional groups that are absent from a particular forest (target community) but predicted to occur in comparable natural forests (reference community).

Despite the recognition that a continuum of biotic patterns related to succession exist, forests should be managed to maintain environmental conditions that support the natural variety and sequence of tree species’ life histories. Each forest tree species can be represented by one of the four types of ecologically invariant life-history trajectories of species turnover: gap colonizers, gap competitors, forest colonizers, or forest competitors. Forest colonizers and forest competitors dominate the climax community, which can regenerate in the same composition over time in the absence of disturbance other than tree deaths due to old age. In contrast, the most shade-intolerant species of gap colonizers and gap competitors depend upon opportunistic disturbances to become established. 

In this review, we touch on the importance of these four types of successional groups, their absence and presence in the community, and how they could be used as a core component to evaluate if the development of the community is progressing towards the restoration of the climatic climax. However, further research in needed to develop the concept of forest succession. This could be undertaken through the inclusion of other biotic components, such as, ground vegetation, wildlife and microorganisms, and their impacts on forest succession as an ecosystem. In closing, we suggest that forests should be managed to maintain environmental conditions that support their natural variety and the sequence of tree species’ life histories.

## Figures and Tables

**Table 1 plants-09-01381-t001:** A simplified framework for the life history dynamics for hemi-boreal tree species in Lithuania. See Appendix A for more details on each species.

Tree Species	Life History Traits						
	Dominant Stand Proportion [40]	Soil Moisture ^A^ [41,42]	Soil Fertility ^B^ [41,42]	Shade Tolerance	Hardiness ^C^ [43]	Life Expectancy [42] (Harvesting Age) [44]	Successional Strategy
**Dominant Forest Tree Species**							
Scots pine (*Pinus sylvestris* L.)	34.6%	1–3 and 5	1–3 and 5	Intolerant	9	300–400 (110)	Disturbance generalist
Norway spruce (*Picea abies* L. Karst)	20.9%	3–4	3–4	Intermediate	7	200–300 (71)	Succession generalist
Silver birch (*Betula pendula* Roth)	22.0%	2–5	2–4	Intolerant	9–10	150 (61)	Disturbance generalist
Black alder (*Alnus glutinosa* L. Gaertn)	7.6%	4–5	3–4	Intermediate	7	180–200 (61)	Disturbance generalist
Grey alder (*Alnus incana* L. Moench)	5.9%	2–5	3–4	Intermediate	9	50–70 (31)	Disturbance generalist
Eurasian aspen (*Populus tremula*)	4.6%	3–4	3–4	Intolerant	9	80–100 (41)	Disturbance generalist
English oak (*Quercus robur* L.)	2.2%	3–4	3–4	Intolerant	6–7	500–600 (121)	Disturbance specialist
European ash (*Fraxinus excelsior* L.)	0.9%	3–5	4–5	Intermediate	7–8	> 300 (101)	Succession specialist
**Other Secondary Native Forest Species**							
Small-leaved lime (*Tilia cordata* Mill.)	0.4%	3	3–4	Intermediate	7	500–600 (61)	Succession specialist
Downy birch (***Betula pubescens* Ehrh)	0.4%	3–5	2–5	Intolerant	9	100 ^D^	Disturbance generalist
European hornbeam (*Carpinus betulus* L.)	0.2%	3	3–4	Tolerant	5	200–300 (61)	Disturbance generalist
Norway maple (*Acer platanoides* L.)	0.2%	3–4	3–5	Tolerant	8	150–300 (101)	Disturbance specialist
White willow (*Salix alba* L.)	<0.2%	4	4–5	Intolerant	8	>100 (31)	Disturbance generalist
Bird cherry (*Prunus padus L.)*	<0.2%	4–5	3–5	Intermediate	9	150 ^D^	Disturbance specialist
Crack willow (*Salix fragilis* L)	<0.2%	4	4–5	Intolerant	8	75 (31)	Disturbance generalist
Field elm (*Ulmus minor* Mill.)	<0.2%	2–4	4	Intermediate	5	300 (101)	Succession specialist
European white elm (*Ulmus laevis* Pall.)	<0.2%	3–4	3–4	Tolerant	6–7	250–300 (101)	Succession specialist
Wych elm (*Ulmus. glabra* Huds.)	<0.2%	3–4	4–5	Tolerant	6	300 (101)	Succession specialist
Wild apple (*Malus sylvestris* L. Mill.)	<0.2%	4–5	3–5	Intolerant	8	300 ^D^	Disturbance specialist
Wild pear (*Pyrus pyraster* L. Burgsd.)	<0.2%	3–4	3–4	Intermediate	6	200–300 ^D^	Disturbance specialist
**Introduced Species**							
European beech (*Fagus sylvatica* L.)	<0.2%	3 [37]	3–4	Tolerant	5	500 (101)	Succession generalist
Sessile oak (*Quercus petraea* Matt. Liebl.)	<0.2%	3	2–3	Intermediate	6–7	500–600 ^D^	Disturbance specialist
Large-leaved lime (*Tilia platyphyllos* Scop.)	<0.2%	3–4	4–5	Intermediate	7	500–600 ^D^	Succession specialist
Wild cherry (*Prunus avium* L.)	<0.02%	3–4	3–4	Tolerant	8	100 ^D^	Disturbance generalist

^A^ Soil moisture is rated on 1–5 scale: 1 = dry and 5 = very wet. ^B^ Soil fertility is rated on a 1–5 scale: 1 = infertile and 5 = very fertile. ^C^ Hardiness refers to the ability of tree to tolerate the cold: 0 = intolerant, 0 °C, and 10 = most tolerant, down to −40 °C [43]. ^D^ Harvest age was not defined.

**Table 2 plants-09-01381-t002:** Successional categories of the hemi-boreal forest tree species establishment and growth in Lithuania [39,41,57,58,59,60,61]. The four types of successional groups (**in bold**) resemble Clark and Clark’s [54] four dominant microsite patterns of tree species (A–D) and Whitmore’s [55] tree species groups (1–4). Modified from Franklin [5].

Growth	Establishment	
	Forest	Gaps
**Forest**	**Forest Competitors or Natural Climax** (A) Advanced self-regeneration under shade and grows best in forest stands; average growth rates, especially as juveniles (1).	**Gap Competitors or Post-pioneers** (C) Regenerates and grows best in gaps, saplings can survive in closed forests; increased juvenile growth potential over groups A or B (3).
	*Tilia cordata* *Tilia platyphyllos* *Fagus sylvatica* *Carpinus betulus*	*Quercus robur* *Quercus petraea* *Fraxinus excelsior* *Ulmus laevis* *Malus sylvestris* *Pyrus pyraster*
**Gaps**	**Forest Colonizers or Pre-climax** (B) Regenerates in shade but shows heightened association with gaps as saplings; growth rates are as low as group A but increase with size (2).	**Gap Colonizers or Pioneers** (D) Regenerates after gap formation and achieves optimal growth at all juvenile stages; juveniles have the highest growth potential (4).
	*Picea abies* *Acer platanoides* *Ulmus glabra* *Ulmus minor*	*Betula pendula* *Betula pubescens* *Pinus sylvestris* *Populus tremula* *Alnus glutinosa* *Alnus incana* *Salix alba* *Salix fragilis* *Prunus avium* *Prunus padus*

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
