# Peer review of "Successional Categorization of European Hemi-boreal Forest Tree Species"

_plants, 2020, doi:10.3390/plants9101381_

Round 1

Reviewer 1 Report

Review of plants-941390: Successional Categorisation of Forest Tree Species

OVERVIEW

This manuscript presents important and valuable ideas about improving forest management.  It also gives a relevant theoretical framework for the ideas.  The goal of the paper is worthy, and the content is important, but the presentation does support the goals effectively.

I reviewed this paper before, and the authors did a good job responding to my comments.  However, the presentation of those ideas still needs considerable improvement.  I now see what the main problem is, and I suggest how to correct it.  The numerous quotations are a structural problem in the manuscript; they clutter the text and hinder communication.  See more comments below

MAIN SUGGESTION

  • Eliminate all quoted text in this paper.
  • There are 52 quotations in the paper: 17 in sections 1 & 2 Background; 22 in the tables; and 13 in section 4 General Suggestions. 
  • Write the main points in your own words. Your ideas will be more clear, more integrated, and more concisely presented if you put it all in your own words and connect all your ideas with your own logic.  (Of course, give references for the main points.) 

OTHER SUGGESTIONS

Abstract

  • One third (1/3) of the Abstract is abstract (“abstract” in a different sense) ideas. Cut this back to one sentence on the integrated forest ecosystem.  You will get more readers if you are more concrete and get to the point quickly.
  • Write directly what you mean to say. For example, in the final sentence why write “target community”?  I guess this means the particular forest you want to manage.  If so, write that.  And why write “reference community”.  I guess that means the comparable natural forest.  “Target” and “reference” have not been defined in the Abstract.  If they are defined in the main text they can used there, after the definition is directly given or somehow strongly inferred.  The reader can figure out what you mean, but you do not want them to have to figure it out; you want them to understand with no effort.
  • If you are not discussing “genetic relativity” in the paper do not mention it at all.

Sections 1 and 2 Background

  • Suggestion: begin your introduction with what is now the final sentence, from Line 75, modified, as for example:

“Forests are systems of interacting organisms; to manage them for tree species composition and production we need thorough knowledge of the variety of tree species' life histories and how they interact”. 

This accurately reflects the title of the paper.  Why not follow up on the title right away?

  • Then go on with some limited theoretical background, but be more concise.
  • Regarding this sentence: “A protection of successional trajectories in the avoidance of habitat shifts can be suggested this far.”  If I understand what you are writing here, I recommend:  “We suggest that forests be managed to maintain environmental conditions that support the natural variety and sequence of tree species’ life histories”.
  • Of course one wants to put one’s work in a theoretical context, but this Introduction over-emphasizes the big picture. You will lose the reader who really does want to know about tree life histories to better manage forests.

Section 4 General Suggestions

  • Again, eliminate quotations. Put all this in your own words adapted especially to your own subject matter here.

Other

  • For “habitat shifts” use “change in environmental conditions”. Or mention change in soil, water, light, as the case may be, and describe these as “habitat shifts”.  I realize my suggestions use more words, but clarity is needed.
  • This paper seems to thoroughly embrace the Clementsian, “super-organism” approach to plant communities. As you know, there is also the Gleasonian argument that the community is merely an assembly of individualistic species (to simplify the idea).  I think the authors should search out the latest ideas on this controversy.  Recent papers have tried to integrate the two points of view.

Reviewer 2 Report

In this manuscript successional status of forest tree species are being described and reviewed. First in general terms and then in a bit more detail for the situation in Lituania. All based on extensive quoting from earlier work. The extensive quoting indeed is review, but the text goes form quote to quote and it is not so clear why the exact quoting is so important. The quoting is from a wide range and age of literature, which by itself is nice. But also, a bit overdone. 

I found the context and also the text confusing, and this is exemplified by the abstract. Line 15-17 are clear and that is what I understand well (“In general, this article discusses the categorisation of forest tree species within the framework of natural ecological climax as an objective of reforestation”.) But the storey start with “genetic relativity and environmental invariance” (l 11) , “giving prominence to invariance”, “invariant continuum of relative scales”...... this is very confusing to me and I see no relation between the classification and species focused treatment...  is this the basis of the classification? Also: “genetic relativity has not been defined and pinned down formally or mathematically yet” (l. 12-13) suggests that the paper is going to do this. But it does not. After reading the abstract twice I still was doubting about the goal and contents of the paper and that remained after reading the whole paper. I just could not make a clear connection, or, maybe better, I did not understand the connection (other than in quite vague terms). I expected genetic relativity and responses to climate and climate change, and towards climax forest. I expected overviews of species scores on criteria so that they could be compared. But I got four classes (by itself fine and okay) and species put into these classes (based on quotes from literature).  There are hints as to the importance of these four classes, the absence and presence of the classes in a community, being the core of the evaluation if the development of the community is going well and towards the climax community. And this is not clearly explained or discussed in the core of the paper.

So, yes the four types are fine, and yes the classification of Lituanian species into these four classes seems okay. But where is the invariance, where the scales, where the criteria, and hardcore data? Where is the elative positioning towards the climax community (why should climax community always be the end goal of restoration? Why the tables are full of quotes, in stead of hardcore criteria and reference for those?

The “general suggestions” section seems a bag of elements to me. Not wrong, but just not clear and leading to a red line, or a goal.

Where are the species interactions that were mentioned as being so important (l. 10,22, 79, ..... 231.)

Also: species have different behaviours at different parts of their range (at different locations on environmental gradients? Or maybe depending of the competing species....)

Also the “concluding remarks” are that way, like the general suggestions. Eg why “all the types of successional trajectories is desirable in every forest community” (line 245)? And what does it mean? It is easy to determine missing classes, but then?

All in all, I found this overall a confusing paper (but the four classes, if filled with criteria and species allocation based on criteria and data, is fine). So there is, in my opinion, much space for improvement.  

Reviewer 3 Report

The work under review is a productive development of the concept of forests as multi-scale multi-species networks, evolving towards the climax as the successional process– pattern of natural regeneration. The article discusses the categorisation of forest tree species within the framework of natural ecological climax as an objective of reforestation. Therefore, the submitted review is important not only for the development of theoretical science, but also for practice –  creation of a new forestry.

I recommend that the authors take into account the scientific results presented in the book “European Russian Forests. Their Current State and Features of Their History. 2017 Editors: Smirnova, Olga V., Bobrovsky, Maxim V., Khanina, Larisa G. (Eds.) https://www.springer.com/gp/book/9789402411713 and

For future work aimed at developing the concep tof forest successionsl development, I recommend that authors turn to other biotic components of forest ecosystems (ground vegetation,animals, microorganisms).

Round 2

Reviewer 1 Report

I have no further comments.

Author Response

Thank you for your support and help to improve our manuscript.

Based on Reviewer 2 we have made minor revision, this included checking our English and adding further clarity throughout the manuscript.

Reviewer 2 Report

I thank the authors for the adaptation of the comments and for the willingness to adapt their story. I see that they have done much to change and that the current version is quite different from the earlier one. The text is much easier to read now, since all the abundant quotes are out and it is more written in their own style and words.

Still, several aspects are rather vague to me. The elements are clear, but the glue in between is not so clear. The way they try to link this to general rules and einsteinian conditions is not so easy to follow. The whole invariance principle means that all is fixed? That species have invariance? But they say themselves that in other locations the position of species is different (and they also have different traits). Is the invariance then the tendency to form climax communities, so a characteristic of the community in stead of form the species? Or is the invariance related to the concept of resilience?

I do see the species categories, and they make sense. But in table 2 they also then mention things in brackets after the species names : is that the category of one of the other systems? (eg Cl&Cl, or Yamamoto, or Chazdon etc.... I think not, to be honest. So what is it then? *(by the way in the category D the in between bracket classifications are in italic and should be normal font). Maybe it would be good to explain the categories of the others, or explain where they take the same group or another and why. Now it is just all “roughly corresponding” to them . What does that mean?

The aim is formulated in a, to me, strange way. To “provide distinction to the concept of invariance”??? And to “support the forest succession to a climatic climax viewpoint”?  I find it difficult to understand what this means and how to find the answer to these aims. The core message to me is that species can be categorized into four classes related to the successional development of the forest, and that the composition can be used to determine the comparison to “old growth” forest under the same conditions. I can also imagine however that some old growth forests types do not have a large variety in species types (eg as is in some old growth beech forests). I also would have welcomed some examples of how to use the comparison, and till what extent the comparison would be valid. Etc. But I guess that is for the future.  

The use of successional trajectories for tree species is weird to me, as mostly the term is used for forest development pathways, and not for individual species (would be species life history strategies)

Unfortunately I did not manage to download the appendices, so I could not evaluate the level of detail.

Concluding: the paper certainly is better to read, while at the same time the concepts are not very clear. Table 1 is very useful, and the categorization of the species in table 2 four classes also. Overall the authors have taken my reservations seriously and adapted the paper.

Author Response

This manuscript is a resubmission of an earlier submission. The following is a list of the peer review reports and author responses from that submission.

Round 1

Reviewer 1 Report

This paper is a valuable contribution.  It gives an intelligent and strong rationale for what basic information we need to manage forests.  It gives a general theoretical background for the need for specific information.  This is a good approach.  However, to convey this message effectively, I recommend extensive re-writing before the paper is published. 

I recommend that the authors reduce the amount of text devoted to abstract ideas and add more concrete examples. 

In "1 Background" the discussion of "power laws" and "scale invariance" needs concrete examples.  Thus "...the forest successional dynamics are driven by an invariant continuum of relative scales, rather than by a variety of characteristic scales..." needs some real-life example.  Likewise the ideas in lines 65-67 need a concrete example.

I think the ideas you presented can be described in simpler terms: Forests are integrated systems of organisms.  For trees, we need thorough knowledge of the variety of tree species' life histories and how species are integrated into the forest system in order to manage forests for stability and production.  Follow this with an example.

The topic of how human affected forests lack certain types of species is important.  Put in here an example with a particular tree species, from your table, showing what the consequences are when this species is lacking.

Again, in "4. General suggestions" I recommend putting in examples.  In previous pages you describe the life history strategies of Lithuanian tree species.  In this section, use that information to give a real life example of how to manage a forest with those species with those life histories to achieve the goals described earlier. 

I think you will get more people to read and cite your paper if you make it more concrete. 

Also, I suggest that you reduce the first two sections of the paper in order to get to sections 3 and 4, with the concrete information on species and management.

Below are some specific questions and points.

I do not understand what you mean by a "potential natural community".  Do you mean a successional community that has not yet reached climax state?

I also do not understand what you mean by "genetic relativity".  I trust this is an important concept, but I am not sure exactly what that concept is.

Also, what do you mean by a "more sophisticated" strategy? How is sophistication rated among regeneration strategies?

Referring to Scots pine, how is it that there is an increase in light and heat under older trees?  We would expect it to be darker and cooler, at least during the day, and maybe warmer at night.

How do you define "forest geneticist"?  I do not think you define it the way it is defined among forestry people I know. 

Reviewer 2 Report

General comments

I understand that this paper discusses the categorization of forest tree species in the framework of climax and that climax could be an objective of forest management and regeneration. However, this discussion appears very theoretical and I feel it few connected with today research. For instance, I agree with the idea developed in L75-77 that “the genetically programmed behaviour of species ensures biotic processes that compensate the unfavourable change”, but the authors do not evoke the vast area of researches which developed and flourishes about plant traits and how those traits vary with environmental conditions. It also presents small reviews over the tree species for forestry in Lithuania from the point of view of their successional categories, but this kind of information figures in many textbooks.

Some specific comments

L45 The concept of invariance while apparently central for the paper is not evoke in the abstract

L63 After “Second…”: it is not an answer to the question raised in L55

L65 There should be an introduction (articulation) before the sentence “As is now becoming …”